# Femora from an exceptionally large population of coeval ornithomimosaurs yield evidence of sexual dimorphism in extinct theropod dinosaurs

Romain Pintore[1,2]*, Raphaël Cornette[3], Alexandra Houssaye[1], Ronan Allain[4]

[1]UMR 7179, Mécanismes Adaptatifs et Évolution (MECADEV), Muséum National d'Histoire Naturelle, CNRS, Paris, France; [2]Structure and Motion Laboratory, Department of Comparative Biomedical Sciences, Royal Veterinary College, Hatfield, United Kingdom; [3]UMR 7205, Institut de Systématique, Évolution, Biodiversité (ISYEB), Muséum National d'Histoire Naturelle, CNRS, Sorbonne Université, EPHE, UA, Paris, France, Paris, France; [4]UMR 7207, Centre de Recherche en Paléontologie - Paris (CR2P), Muséum National d'Histoire Naturelle, CNRS, Sorbonne Université, Paris, France

**Abstract** Sexual dimorphism is challenging to detect among fossils due to a lack of statistical representativeness. The Angeac-Charente Lagerstätte (France) represents a remarkable 'snapshot' from a Berriasian (Early Cretaceous) ecosystem and offers a unique opportunity to study intraspecific variation among a herd of at least 61 coeval ornithomimosaurs. Herein, we investigated the hindlimb variation across the best-preserved specimens from the herd through 3D Geometric Morphometrics and Gaussian Mixture Modeling. Our results based on complete and fragmented femora evidenced a dimorphism characterized by variations in the shaft curvature and the distal epiphysis width. Since the same features vary between sexes among modern avian dinosaurs, crocodilians, and more distant amniotes, we attributed this bimodal variation to sexual dimorphism based on the extant phylogenetic bracketing approach. Documenting sexual dimorphism in fossil dinosaurs allows a better characterization and accounting of intraspecific variations, which is particularly relevant to address ongoing taxonomical and ecological questions relative to dinosaur evolution.

*For correspondence:
romain.pintore@edu.mnhn.fr

Competing interest: The authors declare that no competing interests exist.

## Editor's evaluation

This important contribution offers a convincing analysis of the challenging topic of sexual dimorphism in dinosaurs. Unlike previously published contributions, which are ambiguous, this paper, based on 61 ornithomimosaur fossils, makes a compelling case for measurable differences between male and female individuals. Of particular note, the use of rigorous statistical approaches, a major strength of this manuscript, sets this study apart from previous attempts to tackle this question. Morphological changes are carefully analysed and put into a broader comparative context: the conclusions of this paper allow for interesting comparisons between non-avian dinosaurs and extant groups (e.g., crocodilians, birds, mammals). As such, this manuscript will be of interest to a diverse audience, including palaeontologists, zoologists, and evolutionary biologists.

## Introduction

Dimorphism has been reported in every major dinosaur clade and has often been attributed to sex-specific variation (*Dodson, 1975*; *Chapman et al., 1997*; *Bunce et al., 2003*; *Padian and Horner, 2011*; *Knell and Sampson, 2011*; *Knell et al., 2013*; *Mallon, 2017*; *Saitta et al., 2020*). However, recent studies have demonstrated that most of the documented cases of sexual dimorphism in extinct dinosaurs were most likely biased by ontogenetic changes, taphonomic deformations, and small sample sizes, which substantially affect the representativeness of the inter- and intraspecific diversity and undermine statistical analyses (*Griffin and Nesbitt, 2016*; *Hone et al., 2017*; *Saitta et al., 2020*). For example, a discrete and binary variation between gracile and robust morphologies of femoral bone scars, mostly at the level of the lesser trochanter, has frequently been inferred, with more or less confidence, as sexual dimorphism in various ceratosaurian theropods and non-dinosaurian dinosauri-forms (*Colbert, 1990*; *Raath, 1990*; *Benton et al., 2000*; *Britt and Chure, 2000*; *Piechowski et al., 2014*). More recently, *Griffin and Nesbitt, 2016* demonstrated that this feature no longer appeared dimorphic when accounting for ontogenetic series in the silesaurid *Asilisaurus*. In addition, it has been demonstrated on modern populations that sexual dimorphism could be represented only by very subtle morphological variations, making it even harder to detect in fossils with smaller sample sizes (*Hone et al., 2020*; *Saitta et al., 2020*). At a larger scale, *Mallon, 2017* performed a statistical investigation on a large set of studies that hypothesized sexual dimorphism based on a wide diversity of anatomical proxies across the major clades of non-avian dinosaurs. However, among the 48 described occurrences, only 9 datasets were suitable for statistical test, among which only 1 was considered to rigorously demonstrate dimorphism. Indeed, the combination of a principal component analysis (PCA) and a mixture modeling analysis highlighted that the shift in posterior inclination between the eighth and ninth dermal plates of *Stegosaurus mjosi* was best explained by a bimodal distribution. Yet, there is not robust evidence to postulate that the dimorphism shown in dermal plates would be sex specific (*Saitta, 2015*). As a consequence, it appears that no dataset has yet rigorously demonstrated the presence of sexual dimorphism in non-avian dinosaurs (*Hone et al., 2020*). According to *Mallon, 2017*, one should review three issues when demonstrating sexual dimorphism on extinct organisms: (1) sample size in order to ensure population representativeness; (2) methodology in order to use only suitable analyses to study sexual dimorphism, such as mixture modeling; (3) any other intraspecific morphological variation such as ontogeny and pathology, as well as taphonomy.

Here, we studied the intraspecific femoral variation among a remarkably dense and well preserved population of ornithomimosaurs (*Allain et al., 2014*, *Allain et al., 2022*) from the Angeac-Charente Lagerstätte (Lower Cretaceous of France). *Rozada et al., 2014* and *Rozada et al., 2021* demonstrated that at least 61 ornithomimosaur individuals belonged to the same herd and were deposited in a mass mortality event relying on several evidences (e.g. very limited transport; quality of bone preservation; abundance of individuals with a high skeletal representation preserved in a restricted spatial distribution; catastrophic age profile of the group; deposition of sediment and bones under coeval, poorly oxygenated burial and diagenesis conditions given by their rare earth elements and Yttrium profiles). Thus, the ornithomimosaur herd of Angeac-Charente represents a unique occasion to study

**Table 1.** Number of femora and tibiae from the Angeac-Charente ornithomimosaur discovered between 2010 and 2020.

Minimum Number of Elements (MNEs) and Minimum Number of Individuals (MNIs) are given for each fragmented and complete femora.

|  | Femora | Tibiae |
| --- | --- | --- |
| Left proximal (MNE) | 31 | 31 |
| Right proximal (MNE) | 35 | 35 |
| Left distal (MNE) | 18 | 48 |
| Right distal (MNE) | 22 | 46 |
| Left complete (MNE) | 8 | 13 |
| Right complete (MNE) | 11 | 12 |
| MNI | 46 | 61 |

subtle parameters such as intraspecific variation in extinct dinosaurs. Moreover, the exceptionally high minimal number of individuals among the herd, represented by tibiae and femora (*Table 1*), offers a singular opportunity to test for the presence of dimorphism and characterize its variation. Indeed, and in addition of being the most abundant bones discovered from the Angeac-Charente ornithomimosaur, many hypotheses of sexual dimorphism were formulated based on the hindlimb morphology of non-avian dinosaurs (*Colbert, 1990*; *Raath, 1990*; *Larson, 1994*; *Benton et al., 2000*; *Britt and Chure, 2000*; *Carrano et al., 2002*; *Bunce et al., 2003*; *Piechowski et al., 2014*) but also observed in extant archosaurs (*Schnell et al., 1985*; *Farlow et al., 2005*; *Charuta et al., 2007*; *Bonnan et al., 2008*; *Elzanowski and Louchart, 2022*).

We used a 3D geometric morphometric (3D GM) approach that combines anatomical landmarks and sliding semilandmarks along curves and surfaces on both complete and fragmented femora and tibiae (*Figure 1—figure supplement 1A–B*; *Gunz et al., 2005*; *Gunz and Mitteroecker, 2013*). This method is well suited to study biological objects, including limb bones, and to detect subtle intraspecific shape variations (*Zelditch et al., 2012*; *Botton-Divet et al., 2016*) such as dimorphism (*Fabre et al., 2014*). We then investigated the resulting dataset using PCAs and Gaussian mixture modeling (GMM) as clustering analyses. This clustering analysis calculates the number of Gaussian distributions present in a dataset by maximum likelihood estimations and has been demonstrated as a well-suited method for the identification of dimorphism (*Godfrey et al., 1993*; *Dong, 1997*; *Fabre et al., 2014*; *Manin et al., 2016*; *Mallon, 2017*; *Saitta et al., 2020*).

## Results

We highlight a dimorphic variation in femora from the ornithomimosaur herd of Angeac-Charente (*Figure 1A–B*). This dimorphic variation is localized along the diaphysis (i.e. lateromedial curvature) and toward the distal epiphysis (i.e. lateromedial width) of the femur (*Figure 1C–D*). Distributions along the PC1 of complete femora (28.8%) and distal epiphyses (27.9%) are best described by two clusters with a ratio close to 1:1 according to GMM analyses (see *Supplementary file 1* for details). PC1 scores from both analyses are not significantly correlated to the log centroid size, indicating that size-related effects have no impact on the observed dimorphism (p-value>0.1 for complete femora and distal epiphyses, *Supplementary file 1*).

The most important morphological variation of complete femora is a medial to lateral curvature of the femur (*Figure 1C*). The proximal third of the femur appears deviated toward the lateral side in specimens on the negative part of the PC1 axis, whereas specimens located on the positive part have straight to medially curved femora (*Figure 1C*). Coincidentally, the femoral head is directed medially in the negative cluster while it is inclined ventromedially in the positive one (*Figure 1C*). Regarding distal epiphyses, we selected 6 (out of 10) epiphyses from complete femora because the other 4 were taphonomically altered or pyrite encrusted only in the distal area, which would appear relatively more important in analyses restricted only to this area rather than on the complete morphology (*Supplementary file 2*). Nevertheless, for distal epiphyses, the most important morphological variation along PC1 is the expansion of the lateromedial width relative to the anteroposterior length, which is greater in specimens on the positive part of the PC1 axis than on the negative one (*Figure 1D*). In addition, we highlight that the 6 distal epiphyses from complete femora are consistently attributed to the same clusters between the two analyses (*Figure 1A, B*; *Supplementary file 2*). Hence, our study shows that the straighter the shaft is, the more robust the epiphysis is and that this relationship is dimorphic.

However, there is no robust bimodal distribution on proximal epiphyses, as shown by the GMM analyses (*Figure 1—figure supplement 2*; no consistency in the specimen attribution between complete femora and proximal epiphyses). Similarly, there is no dimorphism in the morphological variation of complete tibiae (*Figure 1—figure supplement 3*) along PC1 (24.1%) and PC2 (20.0%).

## Discussion

The closest extant relatives of non-avian dinosaurs are known to display sexual dimorphism with more or less visibility: birds display variation in their plumage and skeleton (*Schnell et al., 1985*; *Owens and Hartley, 1998*; *Dunn et al., 2001*; *Székely et al., 2007*; *Clarke, 2013*; *Duggan et al., 2015*; *Manin et al., 2016*; *Hone et al., 2017*; *Elzanowski and Louchart, 2022*), whereas the variation is restricted to the skeleton in crocodilians (*Fitch, 1981*; *Farlow et al., 2005*; *Cox et al., 2007*; *Prieto-Marquez*

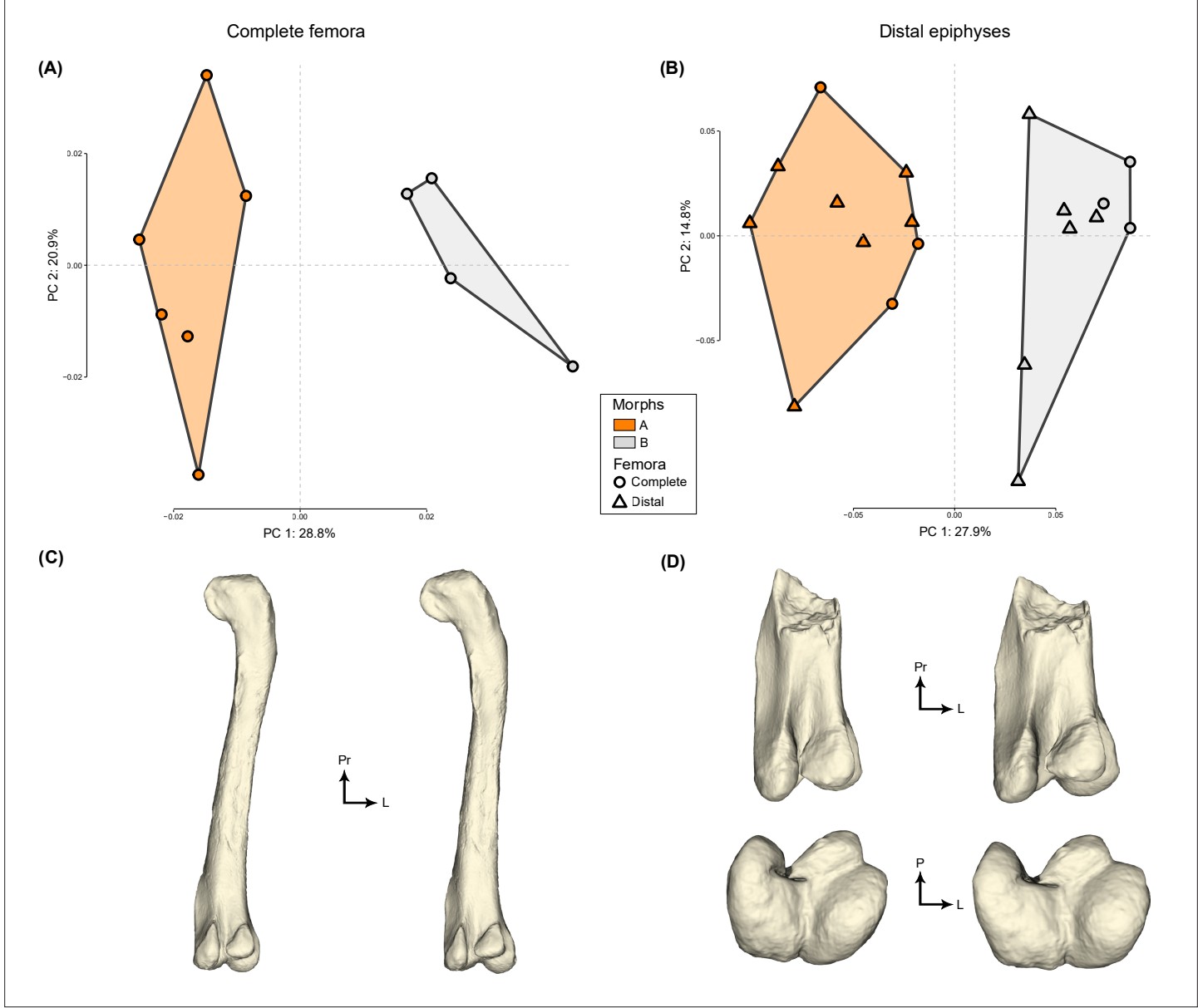

**Figure 1.** The two first axes of the principal component analysis (PCA) for (**A**) complete femora and (**B**) distal epiphyses. Minimal (left) and maximal (right) mean shapes per group for (**C**) complete femora in posterior view and (**D**) distal epiphyses in posterior (top) and distal (bottom) views. Abbreviations: L, lateral; P, posterior; Pr, proximal.

The online version of this article includes the following figure supplement(s) for figure 1:

**Figure supplement 1.** Template of (**A**) right complete femur of ANG10 90 and (**B**) mirrored left distal epiphysis of ANG14 3188 with anatomical landmarks (orange), sliding semilandmarks along curves (dark gray) and surfaces (light gray).

**Figure supplement 2.** The two first axes of the principal component analysis (PCA) for proximal epiphyses of femora.

**Figure supplement 3.** The two first axes of the principal component analysis (PCA) for complete tibiae.

**Figure supplement 4.** Landmark configuration on the templates (**A**) femur; (**B**) tibia, with numerotation following the scheme shown in *Supplementary files 4 and 5*.

**Figure supplement 5.** The two first axes of the principal component analysis (PCA) showing the quantification of the repeatability for the landmark configuration on femora.

**Figure supplement 6.** The two first axes of the principal component analysis (PCA) showing the quantification of the repeatability for the landmark configuration on tibiae.

**Figure supplement 7.** The two first axes of the principal component analysis (PCA) for the original dataset including taphonomically distorted complete femora ANG10 171 (left femur on the plot) and ANG13 2953 (right femur on the plot).

*et al., 2007*; *Bonnan et al., 2008*; *Hone et al., 2017*; *Hone et al., 2020*). The extant phylogenetic bracket (EPB) of non-avian dinosaurs (*Witmer, 1995*) thus implies they were sexually dimorphic too (*Hone et al., 2017*; *Hone et al., 2020*).

A femoral dimorphism of the same nature was demonstrated to be sex-specific among populations of extant tetrapods such as carnivorans and primates. Dimorphism in the femoral obliquity (also termed 'bicondylar angle') was observed in humans, for which females had higher angles than males (*Parsons, 1914*; *Tardieu et al., 2006*; *Hunt et al., 2021*). Moreover, a higher lateromedial width of the distal epiphysis (also termed 'epicondylar width' or 'bicondylar breadth') was demonstrated to vary between sexes in gray wolves and other carnivorans, as well as in primates (*Alunni-Perret et al., 2008*; *Gaikwad and Nikam, 2014*; *Morris and Brandt, 2014*; *Cavaignac et al., 2016*; *Morris and Carrier, 2016*). Whereas no similar sexual dimorphism had been shown – or studied – in non-archosaurian sauropsids to our knowledge, many relevant examples are available in extant and sub-fossil archosaurs. A higher distal width in males than females was demonstrated in wild and captive *Alligator mississippiensis* using linear and geometric morphometrics (*Farlow et al., 2005*; *Bonnan et al., 2008*). *Handley et al., 2016* demonstrated that femoral distal width of the more recently extinct flightless bird *Dromornis stirtoni* was also higher in males than females. To do so, they coupled morphometrics and multivariate statistics with the observation of medullary bone, a sex-specific tissue present in bones of egg-laying female in archosaurians (*Dacke et al., 1993*; *Schweitzer et al., 2005*; *Schweitzer et al., 2007*; *Canoville et al., 2019*). The same kind of sexual dimorphism was observed in modern birds like California gulls (*Larus californicus*; *Schnell et al., 1985*) and in the two extant species of ostriches (*Struthio c. camelus* and *S. c. molybdophanes*) but with reversed proportions between males and females (*Elzanowski and Louchart, 2022*). Furthermore, *Duggan et al., 2015* demonstrated that young male domestic ducks (*Anas platyrhynchos*) had more laterally curved femora than females and that this sexually dimorphic feature disappeared along ontogeny. However, to our knowledge and aside from *Duggan et al., 2015*, data about femoral obliquity is generally unavailable in most studies including sex determination in birds and other sauropsids. Therefore, because the femoral dimorphic features we highlighted in the Angeac-Charente ornithomimosaur herd were also demonstrated to vary between sexes in more or less closely related extant vertebrate clades, we infer it to be sexual.

Ontogenetic allometry was often misinterpreted as sexual dimorphism in archosaurs, as demonstrated in the early dinosauriform *Asilisaurus kongwe*, the crocodylian *A. mississippiensis* and the bird *Rhea americana* (*Griffin and Nesbitt, 2016*; *Hone et al., 2017*; *Hedrick et al., 2022*). However, we found no allometry along the first PC axis (*Supplementary file 1*), which, in addition of rejecting ontogenetic allometry, indicates that the dimorphism is not related to size, as suspected by the homogeneity of femoral lengths highlighted in *Supplementary file 3* among complete femora. Therefore, this indicates no sexual size dimorphism (SSD) in the femur of the Angeac-Charente ornithomimosaurs. SSD is one of the most documented sexual dimorphism across all living organisms, whether it is biased toward females or males (*Darwin, 1874*; *Fairbairn et al., 2007*). There are many examples of observations and/or inferences of SSD and allometric relationships in extant and extinct dinosaurs (*Larson, 1994*; *Bunce et al., 2003*; *Clarke, 2004*; *Székely et al., 2007*; *Remeš and Székely, 2010*; *Olson and Turvey, 2013*; *Handley et al., 2016*; *Manin et al., 2016*; *Fajemilehin, 2017*). However, *Elzanowski and Louchart, 2022* demonstrated that female ostriches had more robust limb bones but smaller average body size than males. This decoupling between size and shape dimorphism is concordant with our results and emphasizes that sexual dimorphism is not necessarily reflected by body size or allometry between limb segments. Thus, size-independent sexual dimorphism should be investigated further extant archosaurs in order to improve inferences about sexual dimorphism in fossils, which are most often represented only by isolated bones.

We did not identify any other dimorphism in either the proximal part of the femur or in complete tibia of the Angeac-Charente ornithomimosaurs (*Figure 1—figure supplements 2 and 3*). However, sexual dimorphism was observed in the proximal ends of femora in extant ostriches (*Charuta et al., 2007*; *Elzanowski and Louchart, 2022*) and California gulls (*Schnell et al., 1985*). In addition, the anteroposterior width of the femoral shaft was demonstrated to vary between sexes among savannah sparrows (*Passerculus sandwichensis*; *Rising, 1987*) and three species of steamer-ducks (*Tachyeres pteneres*, *Tachyeres leucocephalus*, and *Tachyeres patachonicus*; *Livezey and Humphrey, 1984*). Yet, and accordingly with our results, size-independent dimorphism in the avian tibiotarsus seems less

common across the EPB. Indeed, to our knowledge, occurrences of shape dimorphism in the tibia was demonstrated only in California gulls (e.g. width of the shaft; *Schnell et al., 1985*) and in ostriches (e.g. anteroposterior width of the distal epiphysis; only in *Elzanowski and Louchart, 2022* but not in *Charuta et al., 2007*). Furthermore, our observation that sexual dimorphism could be restricted to the femur in the Angeac-Charente ornithomimosaurs and modern archosaurs raises the question of the potential co-variation between the femur and the pelvis. Sexual dimorphism was observed in the ilium of several birds mentioned previously, such as ostriches, steamer-ducks, savannah sparrows, and California gulls (in the antitrochanter width, acetabular width, and synsacrum width and length; *Livezey and Humphrey, 1984*; *Schnell et al., 1985*; *Rising, 1987*; *Charuta et al., 2007*). All measurements were higher in male birds than in female birds except for the width of the ilium, which was higher in female ostriches when measured by *Charuta et al., 2007* but not significantly different between sexes in *Elzanowski and Louchart, 2022*. Additionally, female alligators had a deeper pelvic canal (i.e. distance between the ventral side of the first sacral vertebra and the ventral margin of the ischial symphysis; *Prieto-Marquez et al., 2007*). The dimorphism was located preferably on the femur rather than on the tibia in the Angeac-Charente ornithomimosaur, which suggests that the pelvic area might as well be dimorphic and that seems to be generally the case in some modern avian dinosaurs too (*Livezey and Humphrey, 1984*; *Schnell et al., 1985*; *Rising, 1987*; *Farlow et al., 2005*; *Charuta et al., 2007*; *Prieto-Marquez et al., 2007*; *Bonnan et al., 2008*; *Duggan et al., 2015*; *Elzanowski and Louchart, 2022*). Could the ability to carry eggs restrict the location of sexual dimorphism closer to the hip region? Sexual dimorphism in the pelvic girdle, the proximal hindlimb and the morphological integration between the two in female extant archosaurs should be investigated further to answer this question. However, one would expect that dimorphism in the pelvic morphology would correlate with dimorphism in the proximal instead of the distal femoral portions. Perhaps the shaft curvature toward the lateral side of the femur could have enabled ornithomimosaurs with a wider pelvis (presumably female individuals) to retain their hindlimbs close to the sagittal midline. Nevertheless, the current dataset does not allow to further speculate without the possibility to sex each morphotype and without the integration of femoral with pelvic data.

Our results did not permit to confidently sex each morphotype. Most modern occurrences of femoral sexual dimorphism indicate a wider distal epiphysis among males than females, but *Elzanowski and Louchart, 2022* showed that the opposite was also true for modern and subfossils ostriches. Furthermore, our results indicated that femora with the narrowest distal epiphyses (females in most of modern occurrences) had a laterally deviated shaft. However, *Duggan et al., 2015* demonstrated that only juvenile male Pekin ducks had a laterally deviated shaft, which is not congruent with our results that the widest epiphyses were associated with a straighter morphotype. Paleohistological analyses could enable to verify sex assignment by assessing the presence of medullary bone, as some gravid females may have died during their egg-laying cycle at the time of the mass-mortality event recorded at Angeac-Charente. Indeed, medullary bone was recently demonstrated as probably the most reliable indicator of sex with an extensive distribution across the skeleton (*Canoville et al., 2019*). A paleohistological investigation could also confirm the ontogenetic homogeneity among our femoral sample, as recommended by *Griffin and Nesbitt, 2016*, *Hone et al., 2017* and *Mallon, 2017*.

## Conclusion

Our results demonstrate that the femoral morphology among a large herd of coeval ornithomimosaurs is dimorphic. We identify bimodal distributions along size-independent features that were already reported to vary between sexes in modern archosaurs and other tetrapods (e.g. the width of the distal epiphyses and the lateral deviation of the shaft). Therefore, we infer these features to indicate sexual dimorphism in the Angeac-Charente ornithomimosaurs according to the EPB approach. Our findings inform about the intraspecific variation in non-avian theropods and emphasize the need for description of size-independent dimorphism in modern and closely related taxa with a priori knowledge of the sex. In the future, our results should be completed by paleohistological studies to (1) sex each morphotype and (2) identify the extent of ontogenetic variations within our sample. Additionally, we show that the sex-ratio of the Angeac-Charente ornithomimosaur is close to 1:1 and thus, likely Fisherian (*Fisher, 1930*). It was demonstrated that in extant archosaurs, Fisherian populations are only observed among clutches and hatchlings (*Mayr, 1939*; *Clutton-brock, 1986*; *Liker et al., 2013*) and become generally biased toward females in sub-adult and adult populations, as demonstrated

on crocodilians (*Woodward and Murray, 1993*; *González et al., 2019*) and ratites (*Magige, 2012*; *Prokopenko et al., 2021*). Therefore, paleohistological investigations could help characterize the variation of sex ratio along ontogeny in an extinct dinosaur population and inform if it was truly Fisherian, unlike their extant relatives, or if it also experienced skewness along aging. More broadly, understanding how sex impacted the morphology of an extinct species could shed light on complex evolutionary mechanism such as trade-off between sexually dimorphic features, ecological adaptations, and life-history traits.

## Materials and methods
### Sample and data acquisition
Several complete and fragmented femora and complete tibiae from the Angeac-Charente ornithomimosaur were discovered between 2010 and 2020 (*Table 1*). We removed 158 specimens that were too fragmented and altered by too much oxidized pyrite and trampling (femora: 6 complete, 37 proximal, and 19 distal epiphyses; tibiae: 4 complete, 36 proximal, and 56 distal epiphyses). We selected only fragmented femora that preserved: (1) the most proximal point of the fourth trochanter for proximal epiphyses; (2) the most proximal point of the anteromedial flange for distal epiphyses (*Figure 1—figure supplement 4A*). In total, we digitized 152 specimens (femora: 13 complete, 29 proximal, and 21 distal epiphyses; tibiae: 21 complete, 30 proximal, and 38 distal epiphyses) using the Artec EVA with Artec Studio Professional v. 12.1.1.12 (Artec 3D, Luxembourg, Luxembourg) and the NextEngine with Scan Studio Pro v. 2.0.2 (Next Engine Inc, Santa Monica, United States) for a few specimens (*Supplementary file 3*). After re-examination of digitized specimens, we removed 3 complete femora, 14 proximal and 8 distal epiphyses, and 4 complete tibiae that were distorted (*Figure 1—figure supplement 7*). We thus integrated 10 complete femora, 13 distal and 15 proximal femoral epiphyses, and 17 complete tibiae.

### 3D geometric morphometrics
3D GM is a well-established method for quantifying biological shape variations and has already enabled to identify sexual dimorphism in past studies (*Kaliontzopoulou et al., 2007*; *Cavaignac et al., 2016*). We followed a high-density morphometrics approach using a combination of single anatomical landmarks and sliding semilandmarks along curves and surfaces (*Bookstein, 1997*; *Gunz et al., 2005*). Indeed, most anatomical landmarks are usually concentrated on both ends of limb bones, hence why the use of sliding semilandmarks on the surface was justified on the shaft (*Gunz and Mitteroecker, 2013*; *Botton-Divet et al., 2016*). We digitized 619 landmarks on complete femora (25 anatomical landmarks, 99 sliding semilandmarks on curves, and 495 on surfaces), 479 on proximal (11 anatomical landmarks, 26 sliding semilandmarks on curves, and 442 on surfaces) and distal epiphyses (10 anatomical landmarks, 45 sliding semilandmarks on curves, and 424 on surfaces), and 725 on complete tibiae (23 anatomical landmarks, 219 sliding semilandmarks on curves, and 483 on surfaces; see details in *Figure 1—figure supplement 4*; *Supplementary files 4 and 5*) using the IDAV Landmark software v. 3.0.0.6 (*Wiley et al., 2005*). We digitized anatomical landmarks and sliding semilandmarks along curves on each specimen and sliding semilandmarks along surfaces on one specimen (ANG 10 90), referred to as 'the template' hereafter (*Cornette et al., 2013*). We then automatically projected the sliding semilandmarks along surfaces of the template onto every other specimen following the spline relaxation of semilandmarks along curves using the function 'placePatch' of the Morpho package v. 2.8 (*Schlager, 2017*). Then, we performed 5 iterations of another spline relaxation between landmark configurations of the template and the ones from every other specimen using the function 'relaxLM' of Morpho. Finally, we performed a partial Procrustes fitting in order to compute a Procrustes consensus of every configuration and used it as a target for the 2 last iterations of spline relaxation using the function 'slideLM' of Morpho. These 3 steps of spline relaxations (*Source code 2*) ensured that every semilandmark position was geometrically homogeneous in all specimens (*Gunz et al., 2005*). Finally, we performed a generalized Procrustes analysis (GPA) using the function 'gpagen' of the R package geomorph v. 3.3.1 (*Adams et al., 2013*) in order to align each femur in the Cartesian coordinate system by superimposing them based on their landmark configuration and to rule out the effect of size, location, and orientation of the different landmark configurations (*Gower, 1975*; *Rohlf and Slice, 1990*; *Zelditch et al., 2012*).

## Statistical analyses and clustering

We performed a PCA in order to reduce dimensionalities of the variation and isolate different components of shape variation (*Gunz and Mitteroecker, 2013*). The quantification of repeatability was performed by digitizing landmarks iteratively (n=10) on three morphologically close specimens for complete femora and tibiae, which resulted in 30 configurations for each bone. We then computed a PCA for the two bones (30 configurations each), which showed that all 10 repetitions for each specimen were grouped together and isolated from those of the other specimens along the first two PC axes (*Figure 1—figure supplements 5 and 6* ). This ensured that biological variation was greater than the operator effect, which refers to the ability to reproduce accurately the same landmark configuration multiple times on the same specimen. As recommended by *Mallon, 2017*, we performed mixture modeling analyses without a priori knowledge about the number of groups in order to estimate how many morphological clusters would stand out in our dataset, if any, along each PC axis. Gaussians functions are well suited to describe a biological population, especially when applied to a morphometric dataset (*Baylac et al., 2003*). We used the R package Mclust v. 5.4.7, which calculates the most probable number of clusters in a dataset based on the detection of Gaussian distributions by maximum likelihood estimations (*Scrucca et al., 2016*). Bayesian Information Criteria (BIC; e.g. an approximation of Bayes factors for comparing likelihood) were used to choose which model, among the several ones available, fitted best with our dataset (i.e. the model with the highest BIC), while simultaneously estimating the number of Gaussian distributions (*Fraley and Raftery, 2007*). We computed 3D visualizations that highlighted which feature varied the most along each axis, and between clusters when dimorphism was identified. To do so, we first computed a 3D consensual mesh of all specimens of the sample by using the function 'tps3d' from the R package Morpho v. 2.8 (*Schlager, 2017*) which performed a spline relaxation that minimized the bending energy of a thin plate spline (TPS) between the template landmark configuration and a mean landmark configuration (obtained during the GPA). Then, the function used the resulting TPS deformation to warp the 3D mesh of the template onto the mean shape in order to compute a 3D consensual mesh (*Bardua et al., 2019*). Next, we calculated the mean coordinates of every specimen in each cluster along the PC axis identified as dimorphic by the mixture modeling analysis. Finally, we warped the mean shape, and its associated 3D mesh, onto the mean landmark configurations of each cluster by using the 'shape.predictor' function of geomorph v. 3.3.1 (*Adams et al., 2013*) in order to visualize the 3D shape variation associated with the dimorphic PC axis. We studied the allometry within our sample (i.e. the size-related morphological variation [*Klingenberg, 2016*]), using Pearson's correlation between each PC scores and the log-transformed centroid sizes using the R function 'cor.test.' The code for the totality of these steps is provided in *Source code 1*.

## Acknowledgements

We warmly thank D Augier, J-F Tournepiche (The museum of Angoulême) for access to specimens, A Aumont, G Baron (La Rochelle Museum) for access to specimens in temporary exhibit, F Goussard (MNHN) for 3D digitizations with NextEngine and L Rozada and J Goedert (MNHN) for help with retrieving some specimens. We are indebted to the Audoin & Fils company for sharing their discovery with us and for providing us with logistical support. We would like to thank Mrs. Rodet, the Audoin family and the department of La Charente for donating all the fossil material discovered in Angeac-Charente to the museum of Angoulême. We acknowledge the contribution of professional and amateur paleontologists and the numerous students who participated in the Angeac-Charente excavations since 2010. We also thank B. Bed'hom. (MNHN) for very helpful discussions about sexual dimorphism in extant domestic birds. We thank MJ Benton (University of Bristol) and one anonymous reviewer for contributing to this manuscript through their constructive comments and corrections. We also thank the ERC project GRAVIBONE, the UMR7207 (CR2P) and UMR7205 (ISYEB) for financially supporting the publication fees of this study. Finally, we thank C Bader, A Canoville, C Etienne, J Goedert, R Lefebvre, K Leverger, and C Mallet for constructive discussions and recommendations about digitization, analyses and interpretation of data.

# Additional information

## Funding

| Funder | Grant reference number | Author |
|---|---|---|
| Horizon 2020 Framework Programme | ERC Starting Grant GRAVIBONE 715300 | Alexandra Houssaye |

The funders had no role in study design, data collection and interpretation, or the decision to submit the work for publication.

## Author contributions

Romain Pintore, Resources, Software, Formal analysis, Investigation, Visualization, Methodology, Writing – original draft, Writing – review and editing; Raphaël Cornette, Supervision, Investigation, Methodology, Writing – original draft, Writing – review and editing; Alexandra Houssaye, Conceptualization, Supervision, Funding acquisition, Investigation, Methodology, Writing – original draft, Writing – review and editing; Ronan Allain, Conceptualization, Data curation, Supervision, Funding acquisition, Validation, Methodology, Writing – original draft, Writing – review and editing

## Author ORCIDs

Romain Pintore ⬥ http://orcid.org/0000-0003-2438-614X
Alexandra Houssaye ⬥ http://orcid.org/0000-0001-8789-5545
Ronan Allain ⬥ http://orcid.org/0000-0002-6357-1586

## Decision letter and Author response

Decision letter https://doi.org/10.7554/eLife.83413.sa1
Author response https://doi.org/10.7554/eLife.83413.sa2

# Additional files

## Supplementary files

• Supplementary file 1. Statistical parameters used in this study for size-effect and cluster attribution.

• Supplementary file 2. Cluster attribution for complete femora studied in analyses for both complete femora and distal epiphyses.

• Supplementary file 3. Specimens used in this study. * refers to specimens digitized with the NextEngine, other specimens were digitized using the Artec EVA. Abbreviations: Col. Nb., collection number; F.L., femoral length (maximal distance between proximal and distal epiphyses); L, left; P.W., proximal width; R, right. Specimens are available online on MorphoSource at https://www.morphosource.org/projects/000519447?locale=en.

• Supplementary file 4. Landmark scheme of the femur according to the numerotation shown in Figure S4. Abbreviations: s, anatomical landmarks; c, sliding semilandmarks on curves.

• Supplementary file 5. Landmark scheme of the tibia according to the numerotation shown in Figure S5. Abbreviations: s, anatomical landmarks; c, sliding semilandmarks on curves.

• MDAR checklist

• Source code 1. R script for the various analyses used in this study.

• Source code 2. R script for the different sliding steps used in this study to homogenize the 3D coordinates of semilandmarks along curves and surfaces.

## Data availability

Specimens used in this study are housed in the collections of the Angoulême Museum, Angoulême, France and are available under demand. 3D models used in this study are shared freely and publicly on MorphoSource.

The following dataset was generated:

| Author(s) | Year | Dataset title | Dataset URL | Database and Identifier |
|---|---|---|---|---|
| Pintore R | 2023 | Femora from an exceptionally large population of coeval ornithomimosaurs yield evidence of sexual dimorphism in extinct theropod dinosaurs | https://www.morphosource.org/projects/000519447?locale=en | MorphoSource, 000519447 |

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
