## [Editor Report]

This important contribution offers a convincing analysis of the challenging topic of sexual dimorphism in dinosaurs. Unlike previously published contributions, which are ambiguous, this paper, based on 61 ornithomimosaur fossils, makes a compelling case for measurable differences between male and female individuals. Of particular note, the use of rigorous statistical approaches, a major strength of this manuscript, sets this study apart from previous attempts to tackle this question. Morphological changes are carefully analysed and put into a broader comparative context: the conclusions of this paper allow for interesting comparisons between non-avian dinosaurs and extant groups (e.g., crocodilians, birds, mammals). As such, this manuscript will be of interest to a diverse audience, including palaeontologists, zoologists, and evolutionary biologists.

---

## [Decision Letter]

**Decision letter after peer review:**

Thank you for submitting your article "Femora from an exceptionally large population of coeval ornithomimosaurs yield evidence of sexual dimorphism in extinct theropod dinosaurs" for consideration by *eLife*.

Your article has been reviewed by two peer reviewers, and the evaluation has been overseen by Nizar Ibrahim as Reviewing Editor and Christian Rutz as Senior Editor. The following individual involved in the review of your submission has agreed to reveal their identity: Michael Benton (Reviewer #1).

The reviewers have discussed their reviews with one another, and the Reviewing Editor has drafted this decision letter to help you prepare a revised submission. Only very minor changes/additions are needed.

Essential revisions:

1) Please add a short text section outlining the rationale for selecting the skeletal elements used in this study (femora and tibiae).

2) Please briefly comment on differences in size (or not) between the two groups.

3) Please provide a little more information on taphonomically distorted specimens.

---

## [Author Response]

Essential revisions:1) Please add a short text section outlining the rationale for selecting the skeletal elements used in this study (femora and tibiae).

We addressed this issue and added one sentence near the end of the introduction emphasizing the abundance of hindlimb bones from the Angeac-Charente ornithomimosaur, and the high number of hypotheses of sexual dimorphism formulated on femora (and some on tibiae) among extinct dinosaurs, but also numerous observations from extant archosaurs.

2) Please briefly comment on differences in size (or not) between the two groups.

We added femoral length measurements on complete femora in Table S3 to further emphasize our point that the dimorphism we observed was not linked to body size variations. In addition, we also clarified the sequence of sentence in the Discussion section about ontogenetic allometry and SSD.

3) Please provide a little more information on taphonomically distorted specimens.

When including taphonomically distorted complete femora, dimorphism was still observable, but along the second axis instead of the first axis. We have added the original PCA (PC1 and PC2) in supplementary figure (Figure 8; previously S7) highlighting that taphonomically distorted specimens drove the morphological variation along PC1. We also added a short description in the legend of the PCA explaining why the taphonomically distorted specimens presumably interfered with the biological intraspecific variation.